# Noradrenergic-dependent functions are associated with age-related locus coeruleus signal intensity differences

Kathy Y. Liu [1✉], Rogier A. Kievit [2], Kamen A. Tsvetanov[3,4,5], Matthew J. Betts[6,7,8], Emrah Düzel[6,7,9], James B. Rowe[2,3,5], Cam-CAN*, Robert Howard[1,11] & Dorothea Hämmerer[6,7,8,9,10,11]

The locus coeruleus (LC), the origin of noradrenergic modulation of cognitive and behavioral function, may play an important role healthy ageing and in neurodegenerative conditions. We investigated the functional significance of age-related differences in mean normalized LC signal intensity values (LC-CR) in magnetization-transfer (MT) images from the Cambridge Centre for Ageing and Neuroscience (Cam-CAN) cohort - an open-access, population-based dataset. Using structural equation modelling, we tested the pre-registered hypothesis that putatively noradrenergic (NA)-dependent functions would be more strongly associated with LC-CR in older versus younger adults. A unidimensional model (within which LC-CR related to a single factor representing all cognitive and behavioral measures) was a better fit with the data than the a priori two-factor model (within which LC-CR related to separate NA-dependent and NA-independent factors). Our findings support the concept that age-related reduction of LC structural integrity is associated with impaired cognitive and behavioral function.

[1] Division of Psychiatry, University College London, London, UK. [2] Medical Research Council Cognition and Brain Sciences Unit, University of Cambridge, Cambridge, UK. [3] Cambridge Centre for Ageing and Neuroscience (Cam-CAN), University of Cambridge and MRC Cognition and Brain Sciences Unit, Cambridge, UK. [4] Centre for Speech, Language and the Brain, Department of Psychology, University of Cambridge, Cambridge, UK. [5] Department of Clinical Neurosciences, University of Cambridge, Cambridge, UK. [6] German Center for Neurodegenerative Diseases (DZNE), Magdeburg, Germany. [7] Institute of Cognitive Neurology and Dementia Research, Otto-von-Guericke-University Magdeburg, Magdeburg, Germany. [8] Center for Behavioral Brain Sciences, Magdeburg, Germany. [9] Institute of Cognitive Neuroscience, University College London, London, UK. [10] Wellcome Centre for Human Neuroimaging, London, UK. [11]These authors contributed equally: Robert Howard, Dorothea Hämmerer. *A list of authors and their affiliations appears at the end of the paper. ✉email: kathy.liu@ucl.ac.uk

The locus coeruleus (LC) is a small, elongated nucleus in the rostral pontine brainstem and is the major origin of nora-drenergic (NA) neurons in the central nervous system. The LC–NA system plays an integral role in several autonomic, cognitive, and behavioral functions via connections with widespread areas of the brain and spinal cord[1,2], including wakefulness, pupillary control, anxiogenesis, attention, decision-making, and memory[3]. In vivo T1- or MT-weighted magnetic resonance imaging (MRI) studies have previously associated LC signal intensity (a measure of LC structural integrity) or functional activation with measures of emotional memory[4], emotional regulation/reactivity[5], inhibitory control[6,7], executive function[8,9], sleep quality[10,11], episodic memory[12], heart rate variability (HRV)[13], oddball processing[14], and attentional selectivity[15,16].

Alterations in the LC–NA system, such as specific patterns of LC cell loss and changes in LC MRI signal intensity, have been consistently demonstrated in healthy ageing[17–22] and neurodegenerative disorders such as Parkinson's disease[23] and Alzheimer's disease[24–26]. That these changes are likely to have functional consequences is supported by converging evidence from postmortem and in vivo studies in older adults, which show a positive relationship between measures of LC integrity and a composite measure of cognitive function[27], learning and delayed recall[12], emotional memory for negative events[28], and cognitive reserve[20]. However, the sample sizes from these studies have mainly been small (with one exception of $N = 294$[12]). Given the wide range of cognitive and behavioral functions that are dependent on the LC–NA system, it would be an advantage to consider the relationship between LC integrity, age, and diverse functions within the same cohort and analytical model. This would potentially provide a better understanding of the functional significance of LC–NA integrity differences and improve our understanding of the role of the LC in ageing, and in both presymptomatic and diagnosed neurodegenerative disorders[29].

LC MRI signal intensity is obtained by exploiting the magnetic properties of neuromelanin, a by-product of noradrenaline production. This endogenous contrast agent accumulates inside the noradrenergic neurons of the LC with increasing age, remains stable between 50 and 80 years, and starts to decline by the ninth decade[17,18]. Thus, it is only likely to provide sufficient signal strength to measure in vivo LC structural integrity in older adults[30,31]. Using a large population-based cross-sectional data set of cognitively normal healthy adults ($N = 605$, age range 18–88 years) from the Cambridge Centre for Ageing and Neuroscience (Cam-CAN)[32,33], we previously reported the presence of a quadratic relationship between mean normalized LC MRI signal intensity (LC contrast ratio, or LC CR) and age, the peak occurring around 60 years[22]. A post hoc "two-lines" test provided a breakpoint of 57 years, driven by age-related LC CR decline in the rostral region likely reflecting loss of neuromelanin-containing noradrenergic neurons in the LC. This study analyzes these Cam-CAN participants' cognitive and behavioral outcomes to clarify the functional significance of age-related differences in LC CR.

Assuming that LC MRI signal intensity provides a stronger indirect measure of LC structural integrity in older and not younger adults, we pre-registered our analysis plan (https://aspredicted.org/yf6in.pdf) to test the hypothesis that in older adults (aged over ~60 years, as determined by our previous analyses[22]), LC CR is associated with cognitive and behavioral measures dependent on the NA system. We predict that putatively NA-dependent and NA-independent functions are separable constructs and differentially relate to LC CR, with the former showing a stronger relationship. To test this, we use structural equation modeling (SEM) to compare a second order, multi-dimensional model (where LC CR is an independent, observed variable connecting to two latent NA-dependent and NA-independent factors) (Fig. 1a) to a unidimensional model (where LC CR relates to a single latent variable representing all cognitive and behavioral measures) (Fig. 1b). Cognitive/behavioral factors (measured by variables in the Cam-CAN database) previously demonstrated in vivo to be related to LC–NA system integrity are chosen as NA-dependent variables (emotional memory, emotional reactivity, and response inhibition +/− executive function, sleep quality, and three measures—years of education, verbal intelligence, and occupational complexity—representing cognitive reserve), and those not previously proposed or shown to rely on the LC–NA system are included in the NA-independent factor (general intelligence, face recognition, and sentence comprehension). Each of these pre-registered functions correspond to a task or questionnaire in the Cam-CAN database, apart from emotional regulation and emotional memory, which had multiple outcome measures[32,33], and we justify the choice of the final variables in our analyses in the Methods section.

We predict that a two-factor model would provide the best fit for the data and that older adults would show a stronger relationship between LC CR and the NA-dependent compared with the NA-independent factor. If the a priori two-factor model survives adjustment for age (by inclusion of age as a second independent observed variable relating to LC CR and the latent variables), this would provide strong evidence of model validity. If the two-factor model shows inadequate fit compared with a simpler unidimensional model, we plan to investigate optimal structure with a more exploratory approach.

In the sample ($N = 605$), the two-factor model performs poorly compared with a unidimensional model, particularly in older adults. We discuss potential reasons for this observation, including the impact of age dedifferentiation and the role of NA in attention and arousal. Consistent with our theoretical framework, the relationship between LC CR and cognitive and behavioral measures is significantly stronger in older than younger adults (and within older adults is stronger in rostral versus caudal LC), which remains the case after adjusting for age. This supports the concept that age-related reduction of LC structural integrity is associated with impaired cognitive and behavioral function.

## Results

**The two-factor vs single-factor model.** As the two-factor model showed improved fit after inclusion of executive function, sleep quality, and cognitive reserve variables to the NA-dependent factor, which initially consisted of emotional memory, emotional reactivity, and response inhibition variables ($AIC_{diff} = 7138$, $BIC_{diff} = 7072.5$, $\chi^2_{diff} = 187.62$, $df_{diff} = 55$, $p < 0.001$ for the $\chi^2$ test), these outcome measures were all included in subsequent analyses.

Initial CFA analysis showed that in terms of comparative overall fit, the AIC for the two-factor and single-factor models were equivalent (Table 1). However, according to the BIC, the simpler model with only one single latent factor for LC-related cognitive and behavioral measures (Fig. 1b) provided a better fit than the a priori two-factor model (Fig. 1a) ($BIC_{diff} = 10$), and the likelihood ratio test suggested that the two-factor model did not significantly improve fit compared with the single-factor model ($\chi^2_{diff} = 4.0102$, $df_{diff} = 2$, $p = 0.1346$ for the $\chi^2$ test).

Moreover, although the absolute fit of the two-factor model was acceptable for the data as a whole (Table 1), it failed to converge when a subsequent multiple group analysis was performed with age (younger <57 years or older >57 years) as a group variable with factor loadings constrained to be equal across the two groups. This was due to the estimated standardized

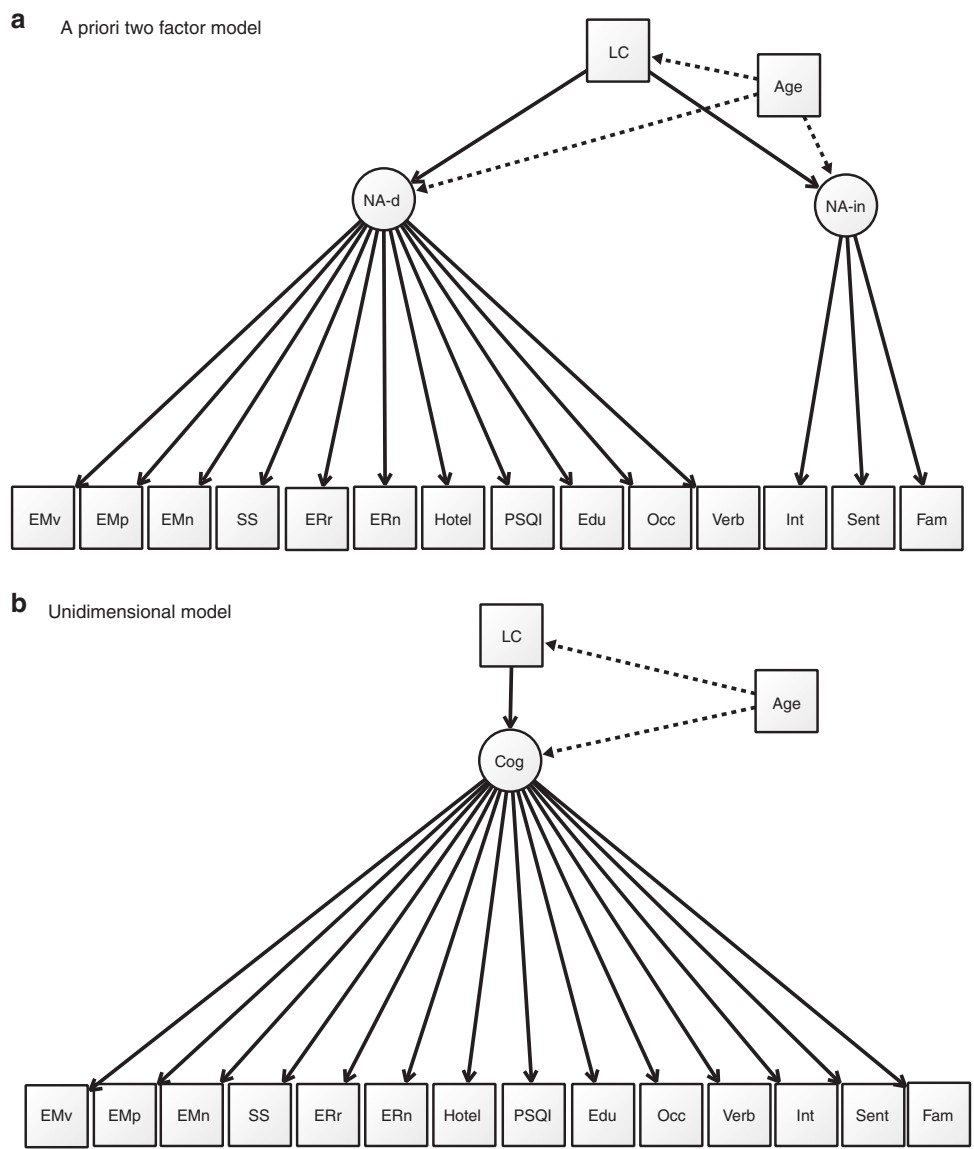

**Fig. 1 The measurement models of LC-dependent function.** LC mean normalized LC signal intensity (LC CR), NA-d NA-dependent factor, NA-in NA_independent factor, Cog single latent factor, EMv emotional source memory—negative valence, EMp emotional implicit memory (priming)—negative valence, EMn emotional object memory—negative valence, SS stop-signal response time (SSRT), ERr emotional reappraisal for negative emotion, ERn negative emotional reactivity, Hotel Hotel task total score, PSQI Pittsburgh Sleep Quality Index total score, Edu years of education, Occ occupational complexity, Verb Spot the Word test as a proxy for verbal intelligence, Int Cattell total score as a proxy for fluid intelligence, Sent sentence comprehension, Fam familiar faces recognition.

| Table 1 Model fits for the two measurement models. | | | | | | | |
|---|---|---|---|---|---|---|---|
| **Model** | **df** | **Chi-square** | ***p*-value** | **RMSEA** | **SRMR** | **AIC** | **BIC** |
| Two-factor | 88 | 237.01 | <0.001 | 0.053 [0.045–0.061] | 0.076 | −14366 | −14168 |
| Single factor | 90 | 241.42 | <0.001 | 0.053 [0.045–0.061] | 0.076 | −14366 | −14177 |

*df degrees of freedom, RMSEA root-mean-square error of approximation, SRMR standardized root-mean-square residual, AIC Akaike information criterion, BIC Bayesian information criterion. For each model, the number of observations was N = 605. As the confirmatory fit index (CFI) was not consistently generated, possibly due to challenging model estimation, we here report only the fit statistics generated for every model. The p-values refer to the chi-squared test (a one-sided test). Source data are provided as a Source Data file.*

covariance between NA-dependent and NA-independent factors exceeding 1 (1.231) in the older group, possibly the consequence of fitting a complex model in a moderately sized subgroup or reflecting the known increase in covariance among different cognitive factors in older adults[34,35]. To facilitate convergence, this correlation was constrained to equal 1, achieving model convergence without further errors, and revealing that the proposed NA-dependent and NA-independent factors were separable in the younger group ($r = 0.388$), but not older ($r = 1.000$) group.

**Table 2 Unidimensional, age-adjusted, multi-group model.**

| Group | Path | Standardized regression estimates | | |
|---|---|---|---|---|
| | | **Overall** | **Rostral** | **Caudal** |
| Younger | LC —> cog (ˆunadjusted for age) | $-0.080$, $z = -0.995$, $p(>|z|) = 0.320$ (ˆ$-0.164$, $z = -2.277$, $p(>|z|) = 0.023$)** | $-0.025$, $z = -0.322$, $p(>|z|) = 0.748$ (ˆ$-0.109$, $z = -1.490$, $p(>|z|) = 0.136$) | $-0.107$, $z = -1.296$, $p(>|z|) = 0.195$ (ˆ$-0.187$, $z = -2.616$, $p(>|z|) = 0.009$)** |
| | Age —> LC | $0.438$, $z = 8.377$, $p(>|z|) < 0.001$** | $0.389$, $z = 7.198$, $p(>|z|) < 0.001$** | $0.439$, $z = 8.556$, $p(>|z|) < 0.001$** |
| | Age —> cog | $-0.260$, $z = -2.357$, $p(>|z|) = 0.018$** | $-0.288$, $z = -2.718$, $p(>|z|) = 0.007$** | $-0.246$, $z = 2.216$, $p(>|z|) = 0.027$** |
| Older | LC —> cog (ˆunadjusted for age) | $0.156$, $z = 2.173$, $p(>|z|) = 0.030$** (ˆ$0.216$, $z = 2.784$, $p(>|z|) = 0.005$)** | $0.194$, $z = 2.468$, $p(>|z|) = 0.014$** (ˆ$0.261$, $z = 3.078$, $p(>|z|) = 0.002$)** | $0.112$, $z = 1.708$, $p(>|z|) = 0.088$ (ˆ$0.160$, $z = 2.200$, $p(>|z|) = 0.028$)** |
| | Age —> LC | $-0.087$, $z = -1.516$, $p(>|z|) = 0.129$ | $-0.102$, $z = -1.771$, $p(>|z|) = 0.077$ | $-0.068$, $z = -1.153$, $p(>|z|) = 0.249$ |
| | Age —> cog | $-0.508$, $z = -5.429$, $p(>|z|) < 0.001$** | $-0.501$, $z = -5.361$, $p(>|z|) < 0.001$** | $-0.515$, $z = -5.492$, $p(>|z|) < 0.001$** |

LC mean normalized LC signal intensity (LC CR), Cog single latent factor.
Regression path estimates between LC CR, the single latent factor, and age for younger and older age groups are shown (age-unadjusted estimates indicated by (ˆ)). Factor loadings were constrained to be equal across the two groups. Rostral and caudal analyses were exploratory and not a pre-registered step. The number of observations was $N = 605$ (269 in the older and 336 in the younger group). The $z$-values represent the Wald statistic (obtained by dividing the parameter value by its standard error), and the $p(>|z|)$ values test the null hypothesis that the parameter equals zero in the population (a two-sided test). Statistically significant ($p(>|z|) < 0.05$) values are indicated by (**). No additional adjustments were made for multiple comparisons. Source data are provided as a Source Data file.

**Older vs younger adults in the single-factor model.** Our pre-registered two-factor model structure did not appear to be compatible for the older group and did not provide a better fit compared with a simpler unidimensional model, our first hypothesis was not supported and our second hypothesis could not be applied to the a priori model. Since we had not anticipated the known increase in covariance between different cognitive factors with age in our pre-registration, we continued to explore the compatibility of the one-factor model (where instead of two factors, a single latent factor represented all the observed cognitive/behavioral variables) (Fig. 1b). The absolute fit of the single latent variable model for the data as a whole was acceptable (Table 1). A multiple group analysis with age as a group variable (younger <57 years or older >57 years) was performed, constraining factor loadings of the cognitive and behavioral variables on the single latent factor to be equal across the two groups to allow comparison of the regression paths of the latent factor to LC CR (and age) across groups (Fig. 1b). Older adults showed a significant positive relationship between LC CR and the single latent factor (standardized regression coefficient $\beta = 0.216$, $z = 2.784$, p(>|z|) = 0.005), which remained significant after adjusting for continuous age as an additional independent variable to the model ($\beta = 0.156$, $z = 2.173$, p(>|z|) = 0.030). In contrast, younger adults showed a significant negative relationship between LC CR and the latent cognitive factor, before ($\beta = -0.164$, $z = -2.277$, p(>|z|) = 0.023), but not after, controlling for age ($\beta = -0.080$, $z = -0.995$, p(>|z|) = 0.320) (Table 2).

Exploratory sub-regional analysis of regression path estimates in the age-adjusted multi-group model revealed a stronger positive relationship between LC CR and the single latent factor in the rostral ($\beta = 0.194$, $z = 2.468$, $p(>|z|) = 0.014$) compared with the caudal ($\beta = 0.112$, $z = 1.708$, $p(>|z|) = 0.088$) LC region in older adults. When we compared the standardized effect sizes between rostral and caudal LC CR on the single latent factor in older adults using Steiger's $z$-test to test for the difference between correlated correlations, the difference between rostral and caudal regression path estimates in older adults was significant ($t = 2.13$, $p < 0.034$ and $t = 2.67$, $p < 0.0081$ for age adjusted and unadjusted models, respectively).

As the difference between significant and not significant is not itself necessarily statistically significant[36,37], we tested for a true difference in the relationship between LC CR and the single latent factor between the two age groups using a likelihood ratio test, which compared a single-factor model where the interaction was constrained to be equal to one where it was freely estimated. This revealed a significant difference in the LC-single latent factor regression path between the older and younger groups in the non-age adjusted ($\chi^2_{diff} = 21.78$, $df_{diff} = 1$, $p = 3.057e-06$ for the

$\chi^2$ test) and age-adjusted model ($\chi^2_{diff} = 5.6894$, $df_{diff} = 1$, $p = 0.017$ for the $\chi^2$ test), a finding that also held when applied to separate rostral and caudal LC regions. However, this was not the case for a control region in the pons (the reference region used to calculate LC CR described in "Methods"), which showed no significant relationship with the single latent factor in older or younger adults and no differences in this relationship between the groups, suggesting that the present results are specific to the LC.

In support of this, the main findings survived the addition of total brain volume (relative to intracranial volume) as an independent observed variable to the age-adjusted unidimensional model. In this exploratory analysis, older adults still showed a significant positive relationship between LC CR and the single latent factor, driven by the rostral region (standardized regression coefficient $\beta = 0.157$, $z = 2.169$, $p(>|z|) = 0.030$ for whole LC, $\beta = 0.192$, $z = 2.465$, $p(>|z|) = 0.014$ for rostral, and $\beta = 0.114$, $z = 1.712$, $p(>|z|) = 0.087$ for caudal LC), and the LC-single latent factor regression paths between older and younger groups remained significantly different. Normalized total brain volume showed a significant negative correlation with age in both groups, and interestingly, was significantly related to the single latent factor in younger ($\beta = 0.157$, $z = 2.169$, $p(>|z|) = 0.030$) but not older adults ($\beta = -0.051$, $z = -0.491$, $p(>|z|) = 0.623$), possibly reflecting age-related differentiation within gray and white matter[38]. The main findings also remained unchanged after the addition of the repetition time (TR) group (30 ms or 50 ms) as a covariate on LC CR.

In the unidimensional, age-adjusted, multi-group model, the single latent factor was significantly associated with all the observed variables except for emotional priming memory for negative valence (EMp), response inhibition (SSRT), emotional reactivity to negatively valenced stimuli (EMn), and sentence comprehension (Sent) variables, although factor loadings varied considerably in strength (Table 3). As expected, apart from SSRT, which had a positive correlation with the single latent factor, variables for which higher values indicate worse function (Hotel task score, PSQI, occupational complexity and sentence comprehension) showed a negative correlation with the single latent factor. In older adults, LC CR explained 4.7% (LC CR and age together explained 30%) of the variance in the single cognitive factor, and in younger adults this was 2.7% (with age 9.2%). This is equivalent to $r = 0.22$ in older and $r = 0.16$ in younger adults, which represents a moderate-to-large effect size[39].

## Discussion
The pre-registered study reveals the relationship between LC CR, age, and multiple cognitive and behavioral functions in an

**Table 3 Unidimensional, age-adjusted, multi-group model.**

| Path | Older (younger) | |
|---|---|---|
| | Standardized regression estimates | $R^2$ |
| Cog —> EMn | 0.744, $z =$ NA, $p(>|z|) =$ NA (0.663, $z =$ NA, $p(>|z|) =$ NA) | 0.553 (0.440) |
| Cog —> EMp | 0.088, $z = 1.136$, $p(>|z|) = 0.256$ (0.063, $z = 1.136$, $p(>|z|) = 0.256$) | 0.008 (0.004) |
| Cog —> EMv | 0.789, $z = 9.422$, $p(>|z|) < 0.001$ (0.634, $z = 9.422$, $p(>|z|) < 0.001$)** | 0.622 (0.402) |
| Cog —> SS | 0.096, $z = 0.522$, $p(>|z|) = 0.602$ (0.069, $z = 0.522$, $p(>|z|) = 0.602$) | 0.009 (0.005) |
| Cog —> ERr | 0.268, $z = 2.417$, $p(>|z|) = 0.016$ (0.209, $z = 2.417$, $p(>|z|) = 0.016$)** | 0.072 (0.044) |
| Cog —> ERn | 0.129, $z = 0.993$, $p(>|z|) = 0.321$ (0.090, $z = 0.993$, $p(>|z|) = 0.321$) | 0.017 (0.008) |
| Cog —> Hotel | −0.323, $z = −4.702$, $p(>|z|) < 0.001$ (−0.243, $z = −4.702$, $p(>|z|) < 0.001$)** | 0.104 (0.059) |
| Cog —> PSQI | −0.186, $z = −2.226$, $p(>|z|) = 0.026$ (−0.122, $z = −2.226$, $p(>|z|) = 0.026$)** | 0.034 (0.015) |
| Cog —> Edu | 0.446, $z = 5.082$, $p(>|z|) < 0.001$ (0.389, $z = 5.082$, $p(>|z|) < 0.001$)** | 0.199 (0.151) |
| Cog —> Occ | −0.198, $z = −1.978$, $p(>|z|) = 0.048$ (−0.113, $z = −1.978$, $p(>|z|) = 0.048$)** | 0.039 (0.013) |
| Cog —> Verb | 0.351, $z = 3.515$, $p(>|z|) < 0.001$ (0.230, $z = 3.515$, $p(>|z|) < 0.001$)** | 0.123 (0.053) |
| Cog —> Int | 0.668, $z = 5.725$, $p(>|z|) < 0.001$ (0.532, $z = 5.275$, $p(>|z|) < 0.001$)** | 0.446 (0.283) |
| Cog —> Sent | −0.041, $z = −0.602$, $p(>|z|) = 0.547$ (−0.028, $z = −0.602$, $p(>|z|) = 0.547$) | 0.002 (0.001) |
| Cog —> Fam | 0.249, $z = 3.295$, $p(>|z|) = 0.001$ (0.192, $z = 3.295$, $p(>|z|) = 0.001$)** | 0.062 (0.037) |

*Cog* single latent factor, *EMv* emotional source memory—negative valence, *EMp* emotional implicit memory (priming)—negative valence, *EMn* emotional object memory—negative valence, *SS* stop signal response time (SSRT), *ERr* emotional regulation negative scale reappraisal, *ERn* emotional regulation reactivity to negative, *Hotel* Hotel task total score, *PSQI* Pittsburgh Sleep Quality Index total score, *Edu* years of education, *Occ* occupational complexity, *Verb* Spot the Word test as a proxy for verbal intelligence, *Int* Cattell total score as a proxy for fluid intelligence, *Sent* sentence comprehension, *Fam* familiar faces recognition.
Regression path estimates and $R^2$ (proportion of variance) for the observed cognitive/behavioral variables in older and younger adults are shown. Factor loadings and variances were constrained to be equal across the two groups. The number of observations was $N = 605$ (269 in the older and 336 in the younger group). Higher measures for SS, Hotel, PSQI, and Occ indicate worse outcomes; thus, they are expected to have negative correlations with LC CR. The $z$-values represent the Wald statistic (obtained by dividing the parameter value by its standard error) and the $p(>|z|)$ values test the null hypothesis that the parameter equals zero in the population (a two-sided test). Statistically significant ($p(>|z|) < 0.05$) values are indicated by (**). No additional adjustments were made for multiple comparisons. Source data are provided as a Source Data file.

integrated statistical model. Compared with previous studies, it uses a lifespan sample with a broad age range (18–88 years) and a large sample size ($N = 605$), with equal representation in each age subgroup. This is important, because of the previously observed quadratic relationship between LC CR and age[22], where division of a sample into older and younger participant groups with narrow age ranges may obscure potential age-related differences between groups and age-related trends in LC CR across the lifespan. Also, the age-continuous cross-sectional sample allowed adjustment for the known linear effect of age on cognition across the lifespan[35].

Although our a priori hypothesized two-factor model converged for the data as a whole, it did not significantly improve fit compared with a simpler unidimensional model. In addition, NA-dependent and NA-independent functions, as represented by their respective observed variables, were separable into two distinct factors in younger but not older adults in our sample. One potential explanation for this could be that older adults show greater "age dedifferentiation", whereby correlations among distinct measures of cognitive function become more intercorrelated with age[40–42], possibly related to neuropathological changes[43]; however, this is not a consistent finding as other studies have also found no evidence for age-related dedifferentiation among cognitive factors[38,44]. A second possible explanation is that although specific cognitive and behavioral functions relying on the LC–NA system have been described, it is conceivable that putatively NA-independent measures such as fluid intelligence, sentence comprehension, and facial recognition might also be reliant on this system due to the widespread distribution of noradrenergic neurons and the role of NA in attention and arousal that underlies diverse tests of cognitive functions. Thus the LC–NA system potentially contributes to all the included observed variables via mainly direct (e.g., emotional memory) or indirect (e.g., via attention) mechanisms.

Consistent with our theoretical framework, we found that compared with younger adults, older adults showed a significantly stronger relationship between LC CR and the single latent factor, while controlling for age, supporting the concept that age-related decline in LC structural integrity due to cell loss has an impact on cognitive function. Sub-regional analyses showed that LC CR remained significantly related to the single latent factor in the rostral, but not caudal region. Although this secondary analysis was exploratory (i.e., not pre-registered), it is consistent with what we would expect from previous post-mortem and in vivo studies in healthy adults, which found greater age-related cell loss[18,45] and LC CR decline[22] associated with impaired cognitive function[12] within the rostral compared with caudal regions of the LC, as well as animal studies, which have indicated that rostral LC cells project to the cerebral cortex and forebrain, whereas caudal LC neurons innervate spinal cord[46,47]. Younger adults, who show age-related increase in LC CR believed to be due to accumulation of neuromelanin inside LC neurons without substantial cell loss, do not show a relationship between LC CR and cognitive/behavioral function after adjusting for age. To strengthen support for the specificity of our findings for the LC, the results were not replicated in a control brain region (pons), or explained by age-related, normalized total brain volume changes.

Consistent with previous in vivo studies in older adults, we found that emotional memory for negatively valenced stimuli[28], as well as three measures (years of education, verbal intelligence, and occupational complexity) that have previously been associated with cognitive reserve[20], were significantly correlated with the single latent variable that showed a positive relationship with LC CR in older adults. Our finding that emotional priming memory for negative valence (EMp) was not related to LC cognitive/behavioral function may be explained by a previous study that showed priming memory for a neutral object paired with an emotional background was not related to valence[48]. Also contrary to our hypothesis, response inhibition (SS) and emotional reactivity to negatively valenced stimuli (EMn) were not significantly related to LC cognitive/behavioral function. As the in vivo studies that formed the basis of these predictions for emotional reactivity[5] and inhibitory control[6,7] were fMRI studies conducted in younger adults, further research is needed to investigate how LC connectivity or reactivity is related to age-related changes in LC structural integrity. For example, a recent study using Cam-CAN data showed age-related differences in the relationship between

response inhibition performance and brain connectivity but not activity in a distributed inhibition network[49], which raises the possibility of compensatory changes in response to age-related reductions in LC structural integrity.

A potential limitation of our study was the nature and degree of missing data. For example, only a small proportion of the whole sample had response inhibition outcomes ($N = 109$), and emotional memory and emotional regulation tasks were mutually exclusive, i.e., participants had completed either emotional memory ($N = 325$) or emotional regulation tasks ($N = 314$), but not both. In addition, despite our large sample, the models tested are relatively complex, which can (and did) affect model estimation. Moreover, we tested a narrow set of models, and alternative models not outside of our initial pre-registered hypotheses might fit the data better. For example, future studies should also explore variables that were not pre-registered in this study, e.g., HRV, episodic memory or alternative measures of cognitive reserve[50]. Although we made pre-registered predictions on the relationship between LC CR and cognitive/behavioral processes, which were based on the wider literature underpinning our hypotheses, we did not pre-register the precise Cam-CAN outcome variables for all processes (i.e., SSRT for response inhibition, Hotel task for executive function, PSQI total score for sleep quality, three measures for negatively valenced emotional memory, and two for emotional reactivity/regulation) that were included in our final analyses. The pre-registered study tested predictions on pre-existing data, which is not as ideal as pre-registering research questions and analysis plans before collection of data. In line with suggested strategies to maximize statistical diagnosticity in these cases[51], we transparently reported in our pre-registered analysis plan that we already had access to the pre-existing data in Cam-CAN, but had not performed any analyses of the cognitive/behavioral measures. Despite the challenges inherent in pre-registering a secondary data analysis, this provides one way to increase the transparency of analysis of pre-existing data, which remains an important tool for exploring research questions[52]. As the LC CR values were obtained from prior work, some limitations described in our previous study will also apply to the current analyses. These include the relatively low resolution of the Cam-CAN MT-weighted images (original voxel size 1.5 mm isotropic), which may have contributed to some of the inter-individual variability of LC CR values, and the fact that the structural imaging protocol used a Repetition Time (TR) of either 30 ms or 50 ms, depending on the participant's SAR estimation. Although our statistical correction for TR differences did not alter the overall findings, it is possible that this may not have entirely accounted for the effect of these differences, which may have influenced the results. While some tasks used in Cam-CAN are standardized (e.g., Cattell test of fluid intelligence), others (e.g, the emotional memory task) were developed specifically for Cam-CAN[32], in order to test specific hypotheses about ageing, and so do not have independent measures of reliability. Despite these limitations, our study emphasizes the importance of further investigating LC MRI signal intensity as a potential biomarker of LC-noradrenergic integrity in older adults, and the value of proposing and testing pre-registered hypotheses. It will be valuable to obtain follow-up investigations of emergent diagnoses of dementia from the Cam-CAN sample, as well as deploy similar models in other large healthy samples such as UK Biobank, using peer-reviewed pre-registration.

In conclusion, this pre-registered study found that age-related differences in normalized LC MRI signal intensity (LC CR), a marker of LC structural integrity, were associated with performance in a range of cognitive and behavioral functions in older but not younger adults. Sub-regional analysis revealed that this effect was localized to the rostral LC. Our findings also suggest that the distinction between NA-dependent and NA-independent functions is confounded by the potential role of NA in attention and arousal that underlies performance on many cognitive tests, and the potential impact of age dedifferentiation. Further studies investigating the LC–NA system in age-related neurodegenerative disease cohorts are warranted.

## Methods

**Participants**. A population-based cohort of healthy adults ($n = 708$) was recruited as part of the Cam-CAN project[32]. Participants were excluded based on several criteria: mini mental state examination (MMSE) < 25; failing to hear a 35 dB 1 kH tone in either ear; poor english language skills (non-native or non-bilingual speakers); self-reported substance abuse and serious health conditions (for example major psychiatric conditions, or a history of stroke or heart conditions); or MRI or MEG contraindications (for example, ferromagnetic metallic implants, pacemakers, or recent surgery). In total, 707 participants were recruited for the cognitive assessment (359 female, age range 18–88 years, mean = 55 years, SD = 18.6), and usable MT-weighted imaging data were collected from 623 individuals (316 female, age range 18–88 years, mean = 54 years, SD = 18.6) (see refs. [32,33] for a detailed study protocol). Ethical approval for the study was obtained from the Cambridgeshire 2 (now East of England—Cambridge Central) Research Ethics Committee (reference: 10/H0308/50), and all participants provided written informed consent prior to the study.

**LC signal intensity values**. For a detailed description of image preprocessing and LC segmentation using Cam-CAN MT-weighted scans, please refer to our previous study[22]. In summary, single-subject MT-weighted images were upsampled to 0.8-mm isotropic resolution, bias-corrected, and spatially normalized to a study-wise space using ANTS v1.2 software package. The LC mask, defined as the conjunction of labeled voxels from two independent manual segmentations on the studywise MT-weighted template, was warped back to the individual native space. Individual scans were inspected for motion artefacts leading to exclusion of 18 participants (age range 20–80 years, mean [SD] = 59 [22] years). Mean LC signal intensity values from the remaining 605 participants (97% of the original sample) were normalized to a reference region (pons) to calculate each individual's contrast ratio (LC CR) that was included in the subsequent analysis. For secondary analyses, each participant's LC was divided into a rostral and caudal region either side of the median voxel along the $z$-axis of the individual's LC to obtain mean rostral and caudal LC CR values for each participant. The structural imaging protocol used a Repetition Time (TR) of either 30 ms or 50 ms, depending on the participant's SAR estimation. As the two TR groups differed significantly in mean age ($t = -2.6$ [523], $p < 0.01$), mean LC CR ($t = 3.8$ [462], $p < 0.001$) and mean reference region signal intensity ($t = -30.8$ [327], $p < 0.001$) using paired $t$ tests[22], we explored the inclusion of TR as a covariate on LC CR in our final model.

**Cognitive and behavioral measures**. Outcome variables from the Cam-CAN database that represented the pre-registered (https://aspredicted.org/yf6in.pdf) cognitive and behavioral factors previously demonstrated in vivo to be related to the LC–NA system (emotional memory[4,28], emotional regulation/reactivity[5], inhibitory control[6,7], executive function[8,9], sleep quality[10,11], and three measures of cognitive reserve[20]) were chosen as NA-dependent variables in the a priori model (Table 4). We chose to select emotional memory and emotional regulation/reactivity measures for negatively valenced stimuli from the Cam-CAN database due to existing evidence supporting a stronger relationship between LC integrity and negatively valenced emotional arousal in older adults[28]. Therefore, emotional memory in our analysis was represented by the sensitivity index (d prime, or d') measures for priming, recognition, and recollection memory for a negatively valenced background image[48]. For emotional regulation and reactivity, we used the negative regulation score (computed by subtracting the negative emotion rating from the negative watch condition from the negative regulate condition) and the negative reactivity score (computed by subtracting the neutral watch score from the negative watch score), respectively[53]. The stop signal response time (SSRT) measures response inhibition during a Stop-Signal/No-Go task and was used to represent inhibitory control[49]. To measure overall executive function, we used the time taken to complete the Hotel task, which tests the ability to plan and multitask[54,55]. Sleep quality data were obtained using the Pittsburgh Sleep Quality Index (PSQI) total score, which incorporates seven subcomponents of sleep quality (subjective sleep quality, sleep latency, sleep duration, sleep efficiency, use of sleep medication, and daytime dysfunction)[56]. Years of education, occupational attainment scores (a scale of 1–10 with 1 indicating the highest occupational complexity), and verbal intelligence (measured using Spot the Word test score[57]) were included to represent cognitive reserve, as these measures were shown to correlate with LC signal intensity in normal ageing when combined into a composite score[20].

Measures not previously proposed or shown to rely on the LC–NA system were included in the NA-independent factor. These were fluid intelligence (measured by

**Table 4 Cognitive and behavioral measures from the Cam-CAN database proposed to rely on either NA-dependent or NA-independent processes (see ref. [32] for a full description of the measures) and included in the SEM analyses.**

| Measure | Description | Outcome variable(s) used in the analysis |
|---|---|---|
| Emotional memory* | Study: view (positive, neutral, or negative) background image, then object image superimposed, and imagine a "story" linking the two; test (incidental): view and identify degraded image of (studied, new) object, then judge memory and confidence for visually intact image of same object, then recall valence and any details of background image from study phase. | For negative valence: priming (accuracy for studied vs. new degraded objects, EMp); familiarity (accuracy for item memory, EMn); recollection (accuracy for background memory, EMv) |
| Emotional regulation* | View (positive, neutral, negative) film clips under instructions to simply "watch" or "reappraise" (attempt to reduce emotional impact by reinterpreting its meaning; for some negative films only), then rate emotional impact (how negative, positive they felt during clip) and the degree to which they successfully reappraised. | Negative regulation (ERr) score: "reappraise" negative vs. "watch" negative ratings and negative reactivity (ERn) score: "watch" negative vs. "watch" neutral negative ratings. |
| Stop-Signal, Go/No-Go* | On Go trials, view a black arrow pointing left or right and indicate the direction of the arrow by pressing left/right buttons. In No-Go trials, participants are required to make no response to a stop signal, which is a red left/right arrow and concurrent tone. | Time needed to inhibit responses on the Stop-Signal trials, also known as Stop-Signal Response time (SSRT). Higher values indicate poorer performance. |
| Hotel Task* | Perform tasks in role of hotel manager: write customer bills, sort money, proofread advert, sort playing cards, alphabetize list of names. Total time must be allocated equally between tasks; there is not enough time to complete any one task. | Time taken to complete task. Higher values indicate poorer performance. |
| Sleep quality* | Complete Pittsburgh Sleep Quality Index (PSQI) which assesses sleep disturbances. | PSQI total score which measures overall sleep disturbance. Higher values indicate worse sleep outcomes. |
| Years of education* | Proposed indicator of cognitive reserve. | Number of years in formal education. |
| Occupational complexity* | Proposed indicator of cognitive reserve. | Score of 1–10 where 10 is least complex. Higher values indicate poorer occupational performance. |
| Verbal intelligence* | Proposed indicator of cognitive reserve. View pairs of items comprising one real word and one invented non-word, and identify the real word. | Spot the Word test score—number of correct answers. |
| Fluid intelligence | Complete nonverbal puzzles involving series completion, classification, matrices, and conditions. | Cattell test score |
| Sentence comprehension | Listen to and judge grammatical acceptability of partial sentences, beginning with an (ambiguous, unambiguous) sentence stem (e.g., Tom noticed that landing planes…) followed by a disambiguating continuation word (e.g., are) in a different voice. Ambiguity is either semantic or syntactic, with empirically determined dominant and subordinate interpretations. | Reaction time, proportion of unacceptable responses in each condition. Higher values indicate poorer performance. |
| Face recognition for familiar faces | View faces of famous people (and some unknown foils), judge whether each is familiar, and if so, what is known about the person (occupation, nationality, origin of fame, etc.), then attempt to provide person's name. | Accuracy (identifying information or full name given) as a proportion of number of faces recognized as familiar, subtracting false alarms (unknown faces given "familiar" response) |

Measures proposed to be NA-dependent are indicated by (*). Measures for which higher values indicate poorer outcomes/performance were predicted to have a negative or inverse correlation with LC CR.

Cattell Culture Fair test total score[58]), sentence comprehension, and familiar faces recognition task performance (Table 4).

**SEM analyses.** SEM is a technique that can model competing hypotheses for the relationship between multiple observed variables and the construct(s) of interest (latent variable(s)). The observed variables are combined to form a composite measure, with the assumption that this will be a more reliable estimate of the construct than any one item on its own[59].

Prior to the SEM analyses, the neural and cognitive/behavioral measures were scaled to lie between 0 and 1 to improve convergence. SEMs were estimated using the Lavaan version 0.6.3[60] package in R version 3.5.1[61]. We assessed overall model fit using the $\chi^2$ test, root-mean-square error of approximation (RMSEA) and its associated confidence interval, and standardized root-mean-square residual (SRMR). Good model fit was defined as: RMSEA < 0.05 (acceptable: 0.05–0.08), SRMR < 0.05 (acceptable: 0.05–0.10). Although the confirmatory fit index (CFI) is also normally reported, it was not consistently generated for a subset of our sample, possibly due to challenging model estimation. Therefore, for consistency, we report here only the fit statistics generated for every model (not because the CFI yielded undesirable results). We used full information maximum likelihood estimation (FIML) to use all available data and the robust maximum likelihood estimator with a Yuan–Bentler scaled test statistic (MLR). Nested models were compared via

Akaike information criterion (AIC), Bayesian information criterion (BIC), and the chi-square difference test. The strength of individual paths was evaluated by reporting standardized parameter estimates (r values of 0.10 were defined as small effects, 0.20 as typical, and 0.30 as large[39]) and tested using a Wald test with an alpha threshold of 0.05.

To investigate whether the covariance structure in our model changed with age, we defined two subgroups by age either side of the breakpoint of 57 years obtained in our previous study[22] to give a younger (18–56 years) and older (57–88 years) subgroup, and employed multiple group analysis using the lavaan package[60]. As exploratory secondary analyses, we compared the model outputs between rostral and caudal LC regions, due to evidence that age-related cell loss disproportionately affects the rostral region, which is believed to have stronger connections to the cerebral cortex and forebrain (including hippocampus) structures. To explore the specificity of our findings to the LC relative to general age-related brain changes, we also explored the model outputs for whole, rostral and caudal LC regions after the addition of total brain volume (relative to intracranial volume) as a third independent observed variable relating to LC CR and the single latent factor (with age relating to total brain volume).

**Reporting summary.** Further information on research design is available in the Nature Research Reporting Summary linked to this article.

## Data availability

The Cam-CAN is an open-access data set available at https://camcan-archive.mrc-cbu.cam.ac.uk//dataaccess/. The Cam-CAN response inhibition (SSRT) data are not yet publicly available. The pre-registered analysis plan can be found at https://aspredicted.org/yf6in.pdf.

## Code availability

The R code that supports the analyses within this paper are available at https://github.com/k-y-liu/SEM-Cam-CAN-LC-code.

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

## Acknowledgements

The views expressed are those of the authors and not necessarily those of the NHS or the Department of Health. K.L is supported by the Medical Research Council (MR/S021418/1). R.H. is supported by the UCLH NIHR BRC. D.H. is supported by the Medical Research Council (Grant Ref: MR/P012698/1), SFB (Sonderforschungsbereich) 1315, Project B6, Research Training Group 2413 'SynAge' TP12, and ARUK SRF2018B-004. M.B. is supported by the Human Brain Project (SP3 WP 3.3.1) and Research Training Group 2413 'SynAge' TP12. K.T. was supported by a British Academy Postdoctoral Fellowship (PF160048). The Cambridge Centre for Ageing and Neuroscience (Cam-CAN) research was supported by the Biotechnology and Biological Sciences Research Council (grant number BB/H008217/1). Additional support to RAK was from the Medical Research Council (grants SUAG/010 RG91365 and SUAG/014 RG91365) and the European Union's Horizon 2020 research and innovation program under grant agreement No 732592. JBR is supported by the Wellcome Trust (103838) and the Medical Research Council (SUAG/051 G101400). We are grateful to the Cam-CAN respondents and their primary care teams in Cambridge for their participation in the Cam-CAN study. We also thank colleagues at the MRC Cognition and Brain Sciences Unit MEG and MRI facilities for their assistance.

## Author contributions

K.L., R.H., D.H., J.B.R., and R.K. conceived the study. K.T. provided response inhibition performance data. K.L. performed the data analysis under guidance from D.H and R.K. and supervision from R.H. R.K., K.T., E.D., M.B., J.B.R., D.H., and R.H. interpreted the results and provided feedback on the paper written by K.L.

## Competing interests

The authors declare no competing interests.

## Additional information

## Cam-CAN

Lorraine K. Tyler[3], Carol Brayne[3], Edward T. Bullmore[3], Andrew C. Calder[3], Rhodri Cusack[3], Tim Dalgleish[3], John Duncan[3], Richard N. Henson[3], Fiona E. Matthews[3], William D. Marslen-Wilson[3], James B. Rowe[3], Meredith A. Shafto[3], Karen Campbell[3], Teresa Cheung[3], Simon Davis[3], Linda Geerligs[3], Rogier Kievit[3], Anna McCarrey[3], Abdur Mustafa[3], Darren Price[3], David Samu[3], Jason R. Taylor[3], Matthias Treder[3], Kamen A. Tsvetanov[3], Janna van Belle[3], Nitin Williams[3], Lauren Bates[3], Tina Emery[3], Sharon Erzinçlioglu[3], Andrew Gadie[3], Sofia Gerbase[3], Stanimira Georgieva[3], Claire Hanley[3], Beth Parkin[3], David Troy[3], Tibor Auer[3], Marta Correia[3], Lu Gao[3], Emma Green[3], Rafael Henriques[3], Jodie Allen[3], Gillian Amery[3], Liana Amunts[3], Anne Barcroft[3], Amanda Castle[3], Cheryl Dias[3], Jonathan Dowrick[3], Melissa Fair[3], Hayley Fisher[3], Anna Goulding[3], Adarsh Grewal[3], Geoff Hale[3], Andrew Hilton[3], Frances Johnson[3], Patricia Johnston[3], Thea Kavanagh-Williamson[3], Magdalena Kwasniewska[3], Alison McMinn[3], Kim Norman[3], Jessica Penrose[3], Fiona Roby[3], Diane Rowland[3], John Sargeant[3], Maggie Squire[3], Beth Stevens[3], Aldabra Stoddart[3], Cheryl Stone[3], Tracy Thompson[3], Ozlem Yazlik[3], Dan Barnes[3], Marie Dixon[3], Jaya Hillman[3], Joanne Mitchell[3] & Laura Villis[3]

