## [Peer Review File · Nature Communications]

Reviewers' Comments:

Reviewer #1:

Remarks to the Author:

Lower density of neurons in locus coeruleus (LC) have been found to correlate with cognitive decline in histology. Neurons in LC contain neuromelanin, which can be delineated in images acquired with a magnetization transfer (MT) preparation pulse. The technical premise of the manuscript is that contrast in the MT images should correspond to LC neuromelanin content (or neuronal density – in older adults at least). The authors use MT contrast as a surrogate for LC neuronal integrity and examine correlations between MT contrast in LC and cognitive measures. The results are convincing and the manuscript is of interest to researchers studying aging, cognition, and cognitive decline.

In particular, the results from the rostral-caudal analysis are of high interest for researchers studying Alzheimer's disease as neuronal loss in Alzheimer's disease is greatest in the rostral portion of LC. The rostral-caudal analysis agrees with this literature as well as with known projections of LC axons and may explain why LC CR is significantly related to cognitive function in the rostral region – i.e. rostral LC axons project to the cerebral cortex and forebrain whereas axons in caudal LC project to the cerebellum and spinal cord [1,2] and, as the authors note, histological work. The significance of this analysis could be improved by expanding the discussion of these results.

[1] doi: 10.1016/0006-8993(78)90861-2

[2] doi: 10.1016/0006-8993(84)90884-9

The authors use MT data from the Cam-CAN project in their analysis. This is an open dataset. There are no significant flaws in imaging processing and the methodology is highly reproducible as the authors map standard space regions of interest to individual subjects. However, the in-plane resolution of the MT/neuromelanin images used is a bit low (1.5 mm x 1.5 mm) for LC delineation and there is variation in TR for this sample – the authors note that identical processing pipelines are used from their previous work to correct these issues (i.e. upsampling the MT images and correcting for TR variation) but this analysis has similar limitations as their earlier analysis and they should probably mention those here as well.

Reviewer #2:

Remarks to the Author:

Major Comments

- In general, I like the idea of fixing true a priori hypotheses prior to running a study. Nevertheless, I have a hard time to consider the present study as a truly pre-registered study. A core idea of preregistration is to define the analysis plan BEFORE the data was seen. This is clearly not the case in the present study. Moreover, a part of the data, i.e., the LC data, is exactly the same set of data that was used to generate some of the hypotheses and analysis parameters, e.g., the group split at ~ 57 years (the preregistration says > 60 years). At least from a typical cross-validated prediction framework, this appears strange. Would one not expect to generate a prediction (peak LC at ~ 57 years) on a „training-set“ and test the derived predictions on a „hold-out set“?

On a related note, the choice of behavioral indicators in each task was not defined in the preregistration (only the type of task that was used). Also, there is no justification for the final indicator choices in the manuscript. Here I do not see, were the preregistration would reduce the „researcher degrees-of-freedom“ beyond what would be achieved in a traditional report / analysis.

- Furthermore, it is unfortunate that the preregistration was not itself peer reviewed. The second

hypothesis, i.e., stronger association between LC and NA-dep tasks in older adults, is completely dependent on the first hypothesis, i.e., NA-dep and NA-indep tasks are separable. In a proper peer reviewed preregistration there would have been the possibility to select for a case-by-case analysis plan, i.e., what happens if hypothesis 1 is not supported.

In the current situation, it rather feels like a dead end. Given that a two-factor solution is not supported, it feels strange to proceed with the preregistered analysis plan for hypothesis 2.

In my opinion, a proper way out would be to run the preregistered CFA analyses as reported and preregistered. Stop there following the preregistration plan, openly stating that hypothesis 1 was not supported and hence hypothesis 2 can not be followed up immediately. Then start an exploratory, data-driven set of analyses, were you uncover the latent structure of your behavioral tasks (e.g., via exploratory factor analysis). If this is done you may exploratorily look for associations with LC signal intensity.

The on-factor solution derived from the pre-specified models appears rather unsatisfactory in the current version of the manuscript. Given the large number of indicators it is not surprising that at least one factor can be extracted. But in this way it is unclear what causes the shared variance between the loading tasks. This can be everything. The fact that some tasks do not really load on this factor might suggest that there is a more subtle, deeper structure – maybe a several factor solution that is not covered by the authors predictions. Understanding this structure might help to reveal a more specific relation to LC signal intensity – that might form the basis for future preregistered studies.

As a general comment: Do not get me wrong here. This reviewer does not consider exploratory analysis as having less scientific value than preregistered analyses. Rather the contrary. Many important scientific discoveries were reached by exploration. Exploration is completely justified (as long as it is clearly marked as such). Both, exploratory and hypothesis driven, confirmatory research have their place in scientific discovery.

- It appears crucial to provide estimates of the reliability of the behavioral measures – given that most of them are derived from experimental tasks (and not from established psychometric test batteries / questionnaires). Understanding the psychometric properties of the set of tasks (especially if it was not yet analyzed as stated in the preregistration) might help to identify potential reasons for model misfit.

Reviewer #3:

Remarks to the Author:

This study employed data from the Cam-CAN database to examine how locus coeruleus structural signal intensity relates to cognitive function. The authors preregistered the hypothesis that, in adults over the age of 57, LC signal intensity would relate to cognitive variables shown previously to be associated with LC and not with apparently LC-independent cognitive variables. SEM modeling indicated that this model did not have a better fit than a single-factor model in which LC signal intensity was generally related to cognition. Older adults showed a significant positive relationship between LC signal intensity and the single cognitive latent variable, a relationship that remained significant after including age in the model. Younger adults showed a significant negative relationship that did not hold up when controlling for age. The relationship in older adults was a moderate to large effect size.

This is an important paper for an emerging literature on the relationship between LC structure and cognition. It is the first to examine a large range of cognitive variables and it has a large sample of older adults. The findings are interesting and suggest that LC integrity contributes to cognition in general in aging. The relationship between LC signal intensity and the cognitive latent factor in older adults was a moderate to large effect size. It could help contextualize this if some other brain

measures were provided as comparisons. For instance, what is the relationship between whole brain volume (relative to intracranial volume) and the cognitive latent factor? Or hippocampal or PFC volume? Understanding where the LC signal intensity ranks relative to these measures of brain structure that previously have been associated with cognition would help orient the reader.

The authors are to be commended for putting together a preregistered plan of analyses. However, it looks like the original main plan only included three variables as NA-dependent tasks (emotional memory, emotional regulation and response inhibition), whereas in the model employed in the paper, a number of other variables were included in the NA-dependent set. This discrepancy should be acknowledged and explained.

p. 3: "In vivo T1- or MT-weighted magnetic resonance imaging (MRI) studies have previously associated LC signal intensity (a measure of LC structural integrity) or functional activation with measures of emotional memory 1, emotional regulation/reactivity 2, inhibitory control 3,4, executive function 5,6 and sleep quality 7,8." - LC structure/function has also been associated with episodic memory (Dahl et al., 2019), heart rate variability (Mather et al., 2017), oddball processing (Murphy et al., 2014) and attentional selectivity (Clewett et al., 2018, Lee et al., 2018).

p. 9: Exploratory analyses revealed the relationship between the single latent cognitive variable and LC signal intensity had a smaller p value for the rostral than the caudal LC region. Was the difference in the rostral and caudal correlations significant?

p. 10: "However, this was not the case for a "control" region in the pons (the reference region used to calculate LC CR described in Methods), which showed the same relationship between cognition and age in older and younger adults, suggesting that the present results are specific to the LC." - was the relationship with the control region significant?

p. 18: It is stated that each participant's LC was divided into a rostral and caudal region either side of the median voxel along the z-axis. Which median voxel? At the individual level (so different median voxels for different participants) or based on some group template?

Mara Mather
(I sign all reviews.)

Reviewers' comments:

Reviewer #1:

Lower density of neurons in locus coeruleus (LC) have been found to correlate with cognitive decline in histology. Neurons in LC contain neuromelanin, which can be delineated in images acquired with a magnetization transfer (MT) preparation pulse. The technical premise of the manuscript is that contrast in the MT images should correspond to LC neuromelanin content (or neuronal density – in older adults at least). The authors use MT contrast as a surrogate for LC neuronal integrity and examine correlations between MT contrast in LC and cognitive measures. The results are convincing and the manuscript is of interest to researchers studying aging, cognition, and cognitive decline.

1. In particular, the results from the rostral-caudal analysis are of high interest for researchers studying Alzheimer's disease as neuronal loss in Alzheimer's disease is greatest in the rostral portion of LC. The rostral-caudal analysis agrees with this literature as well as with known projections of LC axons and may explain why LC CR is significantly related to cognitive function in the rostral region – i.e. rostral LC axons project to the cerebral cortex and forebrain whereas axons in caudal LC project to the cerebellum and spinal cord [1,2] and, as the authors note, histological work. The significance of this analysis could be improved by expanding the discussion of these results.

[1] doi: 10.1016/0006-8993(78)90861-2

[2] doi: 10.1016/0006-8993(84)90884-9

Response 1 R1:

We are grateful to Reviewer #1 for this suggestion. We have now added to the relevant paragraph in our Discussion, "... it is consistent with what we would expect from previous postmortem and in vivo studies in healthy adults, which found greater age-related cell loss ^{10,35} and LC CR decline ¹⁴ associated with impaired cognitive function (Dahl et al. 2019) within the rostral compared with caudal regions of the LC, **as well as animal studies, which have indicated that rostral LC cells project to the cerebral cortex and forebrain, whereas caudal LC neurons innervate spinal cord (Bowden et al. 1978; Westlund et al. 1984).**"

2. The authors use MT data from the Cam-CAN project in their analysis. This is an open dataset. There are no significant flaws in imaging processing and the methodology is highly reproducible as the authors map standard space regions of interest to individual

subjects. However, the in-plane resolution of the MT/neuromelanin images used is a bit low (1.5 mm x 1.5 mm) for LC delineation and there is variation in TR for this sample – the authors note that identical processing pipelines are used from their previous work to correct these issues (i.e. upsampling the MT images and correcting for TR variation) but this analysis has similar limitations as their earlier analysis and they should probably mention those here as well.

Response 1 R2:

We have now included the following additional text in the Methods section, “**The structural imaging protocol used a Repetition Time (TR) of either 30ms or 50ms, depending on the participant’s SAR estimation. As the two TR groups differed significantly in mean age (t = -2.6[523], p<0.01), mean LC CR (t = 3.8[462], p<0.001) and mean reference region signal intensity (t = -30.8[327], p<0.001) (Liu et al. 2018), we explored the inclusion of TR as a covariate on LC CR in our final model.**”

We have also included in the Results section, “**The main findings also remained unchanged after the addition of Repetition Time (TR) group (30ms or 50ms) as a covariate on LC CR.**”

And the Limitations section, “**As the LC CR values were obtained from prior work, some limitations described in our previous study will also apply to the current analyses. These include the relatively low resolution of the Cam-CAN MT-weighted images (original voxel size 1.5mm isotropic), which may have contributed to some of the inter-individual variability of LC CR values, and the fact that the structural imaging protocol used a Repetition Time (TR) of either 30ms or 50ms, depending on the participant’s SAR estimation. Although our statistical correction for TR differences did not alter the overall findings, it is possible that this may not have entirely accounted for the effect of these differences, which may have influenced the results.**”

Reviewer #2:

Major Comments

1. In general, I like the idea of fixing true a priori hypotheses prior to running a study. Nevertheless, I have a hard time to consider the present study as a truly pre-registered study. A core idea of preregistration is to define the analysis plan BEFORE the data was seen. This is clearly not the case in the present study.

Response 1 R2:

We agree with the reviewer that, ideally, pre-registered research questions and analysis plans would be generated prior to collection of data. Testing predictions on preexisting data has been acknowledged to be a practical challenge to the ideals of pre-registration, and it has been suggested that in such cases, statistical diagnosticity can be maximized by pre-registering analysis plans and transparently reporting what was known in advance about the dataset (Nosek et al. 2018). In line with this suggestion, we had openly reported in our pre-registered analysis plan (<http://aspredicted.org/blind.php?x=bh3h8t>) that we already had access to the preexisting data but that no analyses of the cognitive/behavioural measures had taken place. To further increase transparency, we have now also added this information to our manuscript under Limitations, p.17, **“The pre-registered study tested predictions on pre-existing data, which is not as ideal as pre-registering research questions and analysis plans before collection of data. In line with suggested strategies to maximize statistical diagnosticity in these cases (Nosek et al. 2018), we transparently reported in our pre-registered analysis plan that we already had access to the pre-existing data in CamCAN, but had not performed any analyses of the cognitive/behavioural measures.”**

Relatedly, we hope that our explicit statements of where our findings contradict our hypotheses (e.g. page 2, page 18) serve to bolster our transparency.

2. Moreover, a part of the data, i.e., the LC data, is exactly the same set of data that was used to generate some of the hypotheses and analysis parameters, e.g., the group split at ~ 57 years (the preregistration says > 60 years). At least from a typical cross-validated prediction framework, this appears strange. Would one not expect to generate a prediction (peak LC at ~ 57 years) on a „training-set“ and test the derived predictions on a „hold-out set“?

Response 2 R2:

In our pre-registration, we defined older adults as “over ~60 years”, to be consistent with our reports of LC CR peak intensity occurring at “around 60 years” in our previous paper. To improve clarity in the manuscript, we have changed the description of older adults in the Pre-registered hypotheses and analyses section p.5 from “aged over 57 years” to **“aged over ~60 years”**. We agree with the reviewer that it would be important in a future study to cross-validate our findings of age-related differences in LC CR using a different lifespan cohort, and our manuscript emphasises the importance of deploying similar models in other large cohorts in future. However, the objective in the current study was to extend our previous findings of age-related LC CR differences within a

strong theoretical framework by investigating whether the observed LC CR differences were associated with NA-related cognitive/behavioral performance.

3. On a related note, the choice of behavioral indicators in each task was not defined in the preregistration (only the type of task that was used). Also, there is no justification for the final indicator choices in the manuscript. Here I do not see, were the preregistration would reduce the „researcher degrees-of-freedom“ beyond what would be achieved in a traditional report / analysis.

Response 3 R2:

We are grateful to the reviewer for pointing out the discrepancy in the variables between the pre-registered plan and the manuscript. The variables included in the pre-registration were processes chosen based on the wider literature underpinning our hypotheses (described in the manuscript). The precise CamCAN task/measures used to represent these processes were not all named in the pre-registration (i.e. SSRT for response inhibition, Hotel task for executive function, PSQI total score for sleep quality, three measures for negatively valenced emotional memory and two for emotional reactivity/regulation). As explained previously, although we had access to the data and task descriptions when the pre-registration was written, some of the precise outcome variables were not available to us as these were not listed in the published study protocol (Shafto et al. 2014) or on the Cam-CAN database website (<https://camcan-archive.mrc-cbu.cam.ac.uk/dataaccess/>) and we had not yet begun to closely inspect or analyze the data. In retrospect, we have subsequently learned that a second paper (Taylor et al. 2017) did list all the outcome variables, but we had not recognised this when we wrote the pre-registration. We have now acknowledged the discrepancy in our manuscript and hope that our justification for the final variables in the models is now more transparent.

In the Pre-registered hypotheses and analyses section, p.5, we have now reported, **“In our pre-registration, cognitive/behavioral factors (measured by variables in the Cam-CAN database) previously demonstrated *in vivo* to be related to LC-NA system integrity were chosen as NA-dependent variables (emotional memory, emotional reactivity and response inhibition +/- executive function, sleep quality and three measures - years of education, verbal intelligence and occupational complexity - representing cognitive reserve), and those not previously proposed or shown to rely on the LC-NA system were included in the NA-independent factor (general intelligence, face recognition and sentence comprehension). Each of these pre-registered functions corresponded to a task or questionnaire in the Cam-CAN database, apart from emotional regulation and emotional memory,**

which had multiple outcome measures ^{29,30}, and we have justified the choice of the final variables in our analyses in the Methods section. ”

In the Methods section, p.20, “**Outcome variables from the Cam-CAN database that represented the pre-registered cognitive and behavioral factors previously demonstrated in vivo to be related to LC-NA system integrity (emotional memory ^{1,25}, emotional regulation/reactivity ², inhibitory control ^{3,4}, executive function ^{5,6}, sleep quality ^{7,8} and cognitive reserve ¹⁷) were chosen as NA-dependent variables in the *a priori* model (Table 4).We chose to select...**”

Lastly, in the Limitations section, p.17, we have now reported, “**Although we made pre-registered predictions on the relationship between LC CR and cognitive/behavioral processes, which were based on the wider literature underpinning our hypotheses, we did not pre-register the precise Cam-CAN outcome variables for all processes (i.e. SSRT for response inhibition, Hotel task for executive function, PSQI total score for sleep quality, three measures for negatively valenced emotional memory and two for emotional reactivity/regulation).**”

4. Furthermore, it is unfortunate that the preregistration was not itself peer reviewed. The second hypothesis, i.e., stronger association between LC and NA-dep tasks in older adults, is completely dependent on the first hypothesis, i.e., NA-dep and NA-indep tasks are separable. In a proper peer reviewed preregistration there would have been the possibility to select for a case-by-case analysis plan, i.e., what happens if hypothesis 1 is not supported.

Response 4 R2:

We are grateful to Reviewer #2 for suggesting peer-review of pre-registration, which we will consider for future studies. Unreviewed pre-registrations like this study may usefully precede a registered replication (van 't Veer and Giner-Sorolla 2016), and we have added the following statement in our manuscript, “It will be valuable to ... deploy similar models in other large healthy samples such as UK Biobank, **using peer-reviewed pre-registration.**”

We note that even journals that favour formats such as registered reports (e.g. Cortex) generally do not allow pre-registrations for pre-existing datasets given the challenges inherent in doing so. We hope that our explicit statements of where our findings contradict our hypotheses (e.g. page 2, page 18) serve to bolster evidence of our transparency. We also now provide a reference in our Discussion to a novel resource with recommendations for pre-registration of secondary datasets, “**Despite the**

challenges inherent in pre-registering a secondary data analysis, this provides one way to increase the transparency of analysis of pre-existing data, which remains an important tool for exploring research questions (Weston et al. 2019)”.

5. In the current situation, it rather feels like a dead end. Given that a two-factor solution is not supported, it feels strange to proceed with the preregistered analysis plan for hypothesis 2. In my opinion, a proper way out would be to run the preregistered CFA analyses as reported and preregistered. Stop there following the preregistration plan, openly stating that hypothesis 1 was not supported and hence hypothesis 2 can not be followed up immediately. Then start an exploratory, data-driven set of analyses, were you uncover the latent structure of your behavioral tasks (e.g., via exploratory factor analysis). If this is done you may exploratorily look for associations with LC signal intensity. The on-factor solution derived from the pre-specified models appears rather unsatisfactory in the current version of the manuscript. Given the large number of indicators it is not surprising that at least one factor can be extracted. But in this way it is unclear what causes the shared variance between the loading tasks. This can be everything. The fact that some tasks do not really load on this factor might suggest that there is a more subtle, deeper structure – maybe a several factor solution that is not covered by the authors predictions. Understanding this structure might help to reveal a more specific relation to LC signal intensity – that might form the basis for future preregistered studies.

As a general comment: Do not get me wrong here. This reviewer does not consider exploratory analysis as having less scientific value than preregistered analyses. Rather the contrary. Many important scientific discoveries were reached by exploration. Exploration is completely justified (as long as it is clearly marked as such). Both, exploratory and hypothesis driven, confirmatory research have their place in scientific discovery.

Response 5 R2:

Whilst we agree with Reviewer #2 that it would be interesting to complete a data-driven, exploratory factor analysis, we feel that this would be more relevant in a study where no *a priori* hypotheses on the relationships between factors and variables have been made, which is not the case in the current study. Here, we described and extended a strong theoretical framework, within which we initially hypothesized that LC CR would be associated with NA-dependent cognitive processes and that this relationship would be stronger in older vs younger adults. We found that hypothesis 1 was not supported, i.e. NA-dependent and NA-independent tasks were not separable in older adults, which is consistent with the literature describing age dedifferentiation, and in hindsight, this was a concept we ideally could have anticipated in our pre-registration. Thus, our

subsequent exploratory analysis of the one-factor model can be interpreted in the context of relevant literature and is underpinned by strong underlying theory, which gives relevance to any planned statistical tests via pre-registration (Szollosi et al. 2019).

Nonetheless, we agree that the description of our adherence to the pre-registration could be made clearer and have now explicitly stated in our Results, “As our pre-registered two-factor model structure did not appear to be compatible for the older group and did not provide a better fit compared to a simpler unidimensional model, **our first hypothesis was not supported and our second hypothesis could not be applied to the *a priori* model. Since we had not anticipated the known increase in covariance between different cognitive factors with age in our pre-registration, we continued to explore the compatibility of the one factor model...**”. We have also clarified that our second hypothesis was not fully supported by replacing the word ‘hypothesis’ with ‘**theoretical framework**’ in the Abstract and Discussion p.16.

It would also be interesting to conduct an exploratory analysis using all available cognitive and behavioural Cam-CAN measures in a future study.

6. It appears crucial to provide estimates of the reliability of the behavioral measures – given that most of them are derived from experimental tasks (and not from established psychometric test batteries / questionnaires). Understanding the psychometric properties of the set of tasks (especially if it was not yet analyzed as stated in the preregistration) might help to indentify potential reasons for model misfit.

Response 6 R2:

While some tasks used in Cam-CAN are standardized (e.g. Cattell test of fluid intelligence), it is true that others were developed specifically for CamCAN, in order to test specific hypotheses about ageing (e.g, the emotional memory task; see Shafto et al., 2014, for further discussion), and so do not have independent measures of reliability. We now state this limitation in the Discussion, “**While some tasks used in Cam-CAN are standardized (e.g. Cattell test of fluid intelligence), others (e.g, the emotional memory task) were developed specifically for Cam-CAN (Shafto et al. 2014), in order to test specific hypotheses about ageing, and so do not have independent measures of reliability.**”

Reviewer #3 (Remarks to the Author):

This study employed data from the Cam-CAN database to examine how locus coeruleus structural signal intensity relates to cognitive function. The authors preregistered the hypothesis that, in adults over the age of 57, LC signal intensity would relate to cognitive variables shown previously to be associated with LC and not with apparently LC-independent cognitive variables. SEM modeling indicated that this model did not have a better fit than a single-factor model in which LC signal intensity was generally related to cognition. Older adults showed a significant positive relationship between LC signal intensity and the single cognitive latent variable, a relationship that remained significant after including age in the model. Younger adults showed a significant negative relationship that did not hold up when controlling for age. The relationship in older adults was a moderate to large effect size.

1. This is an important paper for an emerging literature on the relationship between LC structure and cognition. It is the first to examine a large range of cognitive variables and it has a large sample of older adults. The findings are interesting and suggest that LC integrity contributes to cognition in general in aging. The relationship between LC signal intensity and the cognitive latent factor in older adults was a moderate to large effect size. It could help contextualize this if some other brain measures were provided as comparisons. For instance, what is the relationship between whole brain volume (relative to intracranial volume) and the cognitive latent factor? Or hippocampal or PFC volume? Understanding where the LC signal intensity ranks relative to these measures of brain structure that previously have been associated with cognition would help orient the reader.

Response 1 R3:

We agree with Reviewer #3 that it would be interesting to examine the relationship between other brain measures, LC CR and cognitive performance. Relatedly, some of the co-authors aim to examine the relationship between LC CR and regional brain volumes in a separate study. We have now explored the model outputs for whole, rostral and caudal LC after the addition of total brain volume (relative to intracranial volume) as an additional independent observed variable, and found that LC CR (especially rostral) was still significantly related to the cognitive factor, with a significant difference between younger and older adults. Interestingly, normalized total brain volume was related to the cognitive factor in younger but not older adults. This may be related to age differentiation, leading to uncoupling between structural brain measures and cognition in older adults. Also, the single latent factor may not have been an optimal or validated measure of 'general' cognition, as it was derived from specific cognitive variables that were selected on the basis of being putatively NA- dependent or independent and related (or unrelated) to LC CR, and did not include over half the available cognitive measures in the Cam-CAN dataset (<https://camcan-archive.mrc->

cbu.cam.ac.uk/dataaccess/). Therefore, the effect of LC CR on the single latent factor relative to other brain regions previously associated with cognition may give a biased 'ranking' of LC CR. Secondly, there is evidence that distinct age-related differences occur within specific brain regions, for example, PFC (Kievit et al. 2014), where a unifactorial model of ageing for this region was shown to be an oversimplification. This suggests that the comparison of LC CR (where we have observed specific age-related differences) with measures of brain volume on the single latent factor may not be equivalent.

We have now added to the Methods, **“To explore the specificity of our findings to the LC relative to general age-related brain changes, we also explored the model outputs for whole, rostral and caudal LC regions after the addition of total brain volume (relative to intracranial volume) as a third independent observed variable relating to LC CR and the single latent factor (with age relating to total brain volume).”**

In our Results, **In support of this, the main findings survived the addition of total brain volume (relative to intracranial volume) as an independent observed variable to the age-adjusted unidimensional model. In this exploratory analysis, older adults still showed a significant positive relationship between LC CR and the single latent factor, driven by the rostral region (standardized regression coefficient $\beta = 0.157$, $z = 2.169$, $p = 0.030$ for whole LC, $\beta = 0.192$, $z = 2.465$, $p = 0.014$ for rostral, and $\beta = 0.114$, $z = 1.712$, $p = 0.087$ for caudal LC), and the LC-single latent factor regression paths between older and younger groups remained significantly different. Normalized total brain volume showed a significant negative correlation with age in both groups, and interestingly, was significantly related to the single latent factor in younger ($\beta = 0.157$, $z = 2.169$, $p = 0.030$) but not older adults ($\beta = -0.051$, $z = -0.491$, $p = 0.623$), possibly reflecting age-related differentiation within grey and white matter (de Mooij et al. 2018).”**

In the Discussion, **“To strengthen support for the specificity of our findings to the LC, the results were not replicated in a control brain region (pons), or explained by age-related, normalized total brain volume changes.”**

2. The authors are to be commended for putting together a preregistered plan of analyses. However, it looks like the original main plan only included three variables as NA-dependent tasks (emotional memory, emotional regulation and response inhibition), whereas in the model employed in the paper, a number of other variables were included in the NA-dependent set. This discrepancy should be acknowledged and explained.

Response 2 R3:

We are grateful to Reviewer #3 for pointing out the discrepancy in the variables between the pre-registered plan and the manuscript. As explained earlier in Response 3 Reviewer#2, we have now explained the discrepancy and clarified how we chose the final variables in the models. We have also explained in the Results section that the additional putatively NA-dependent variables (Hotel task, sleep and cognitive reserve) were included in the final model because these resulted in improved model fit. We will also update the R code on Github to include this analysis.

In the Pre-registered hypotheses and analyses section, p.5, we have now reported, **“In our pre-registration, cognitive/behavioral factors (measured by variables in the Cam-CAN database) previously demonstrated *in vivo* to be related to LC-NA system integrity were chosen as NA-dependent variables (emotional memory, emotional reactivity and response inhibition +/- executive function, sleep quality and three measures - years of education, verbal intelligence and occupational complexity - representing cognitive reserve), and those not previously proposed or shown to rely on the LC-NA system were included in the NA-independent factor (general intelligence, face recognition and sentence comprehension). Each of these pre-registered functions corresponded to a task or questionnaire in the Cam-CAN database, apart from emotional regulation and emotional memory, which had multiple outcome measures ^{29,30}, and we have justified the choice of the final variables in our analyses in the Methods section. ”**

In the Methods section, p.20, **“Outcome variables from the Cam-CAN database that reflected cognitive processes previously demonstrated *in vivo* to be related to LC-NA system integrity (emotional memory ^{1,21}, emotional regulation/reactivity ², inhibitory control ^{3,4}, executive function and decision-making ^{5,6}, sleep quality ^{7,8} and cognitive reserve ¹²) were chosen as NA-dependent variables in the *a priori* model (Table 4). We chose to select...”**

In the Results section, **“As the two-factor model showed improved fit after inclusion of executive function, sleep quality and cognitive reserve variables to the NA-dependent factor, which initially consisted of emotional memory, emotional reactivity and response inhibition variables ($AIC_{diff} = 7138$, $BIC_{diff} = 7072.5$, $\chi^2_{diff} = 187.62$, $df_{diff} = 55$, $p < 0.001$), these outcome measures were all included in subsequent analyses.”**

Lastly, in the Limitations section, p.17, we have now reported, **“Although we made pre-registered predictions on the relationship between LC CR and cognitive/behavioral processes, which were based on the wider literature**

underpinning our hypotheses, we did not pre-register the precise Cam-CAN outcome variables for all processes (i.e. SSRT for response inhibition, Hotel task for executive function, PSQI total score for sleep quality, three measures for negatively valenced emotional memory and two for emotional reactivity/regulation).”

3. p. 3: *“In vivo T1- or MT-weighted magnetic resonance imaging (MRI) studies have previously associated LC signal intensity (a measure of LC structural integrity) or functional activation with measures of emotional memory ¹, emotional regulation/reactivity ², inhibitory control ^{3,4}, executive function ^{5,6} and sleep quality ^{7,8}.” - LC structure/function has also been associated with episodic memory (Dahl et al., 2019), heart rate variability (Mather et al., 2017), oddball processing (Murphy et al., 2014) and attentional selectivity (Clewett et al., 2018, Lee et al., 2018).*

Response 3 R3:

We are grateful to Reviewer #3 for prompting us to include these relevant studies, which have now been added to the manuscript, “...with measures of emotional memory ¹, emotional regulation/reactivity ², inhibitory control ^{3,4}, executive function ^{5,6}, sleep quality ^{7,8}, **episodic memory** (Dahl et al. 2019), **heart rate variability** (Mather et al. 2017), **oddball processing** (Murphy et al. 2014) and **attentional selectivity** (Clewett et al. 2018; Lee et al. 2018).”

We have also added the following limitation to our Discussion, “...**For example, future studies should also explore variables that were not pre-registered in this study, e.g. HRV, episodic memory and alternative measures of cognitive reserve** (Chan et al. 2018).”

4. p. 9: *Exploratory analyses revealed the relationship between the single latent cognitive variable and LC signal intensity had a smaller p value for the rostral than the caudal LC region. Was the difference in the rostral and caudal correlations significant?*

Response 4 R3:

We are grateful to Reviewer #3 for suggesting this additional analysis. When we compared the standardized effect sizes between rostral and caudal LC CR on the single cognitive factor in older adults using Steiger’s z-test to test for the difference between correlated correlations, we found that the difference between the rostral and caudal regression path estimates in older adults was significant (t=2.13, p<0.034 and t=2.67, p<0.0081 for age adjusted and unadjusted models respectively). We have now added this result in the manuscript as below and will update the R code on Github accordingly.

In the Results section: **“When we compared the standardized effect sizes between rostral and caudal LC CR on the single cognitive factor in older adults using Steiger’s z-test to test for the difference between correlated correlations, the difference between rostral and caudal regression path estimates in older adults was significant (t=2.13, p<0.034 and t=2.67, p<0.0081 for age adjusted and unadjusted models respectively).”**

5. p. 10: *“However, this was not the case for a “control” region in the pons (the reference region used to calculate LC CR described in Methods), which showed the same relationship between cognition and age in older and younger adults, suggesting that the present results are specific to the LC.” - was the relationship with the control region significant?*

Response 5 R3:

No, it was not significant and we have added in our Results, **“...which showed no significant relationship with the single cognitive factor in older or younger adults and no differences in this relationship between the groups..”**

We will also update the R code on Github to include this analysis.

6. p. 18: *It is stated that each participant’s LC was divided into a rostral and caudal region either side of the median voxel along the z-axis. Which median voxel? At the individual level (so different median voxels for different participants) or based on some group template?*

*Mara Mather
(I sign all reviews.)*

Response 6 R3:

Each participant’s LC was divided into a rostral and caudal region either side of the median voxel along the z-axis of the individual’s LC. Their LC was defined after the segmented LC mask from the studywise group template was warped back to individual native space, as described in our previous paper.

We have improved the wording of this sentence to improve clarity, **“...each participant’s LC was divided into a rostral and caudal region either side of the median voxel along the z-axis of the individual’s LC to obtain mean rostral and caudal LC CR values for each participant.”**

Reviewers' Comments:

Reviewer #2:

Remarks to the Author:

I am still not fully convinced with some of the authors responses to my comments. Hence, I would suggest to at least drop "a pre-registered cross-sectional study" from the title. But whether the author's approach to pre-registration for an analysis plan for pre-existing data is sound, is a matter of debate - and should therefore be left to the field.

In addition, I appreciate the authors transparent reporting and acknowledge that the data and results are potentially interesting for the field. Hence, I do not have any further issues or concerns.

Reviewer #3:

Remarks to the Author:

The authors have addressed the concerns raised in the previous round and I do not have further comments to add.

Mara Mather